# Parenting Styles and Self-Esteem in Adolescent Cybervictims and Cyberaggressors: Self-Esteem as a Mediator Variable

**DOI:** 10.3390/children9121795

**Published:** 2022-11-23

**Authors:** Maite Garaigordobil, Raúl Navarro

**Affiliations:** 1Faculty of Psychology, University of the Basque Country, 48940 Leioa, Spain; 2Faculty of Education and Humanities, University of Castilla-La Mancha, 16071 Cuenca, Spain

**Keywords:** cyberbullying, self-esteem, parenting styles, adolescence, violence

## Abstract

Background: Family relationships and self-esteem are relevant variables into the understanding of cyberbullying. However, little is known about the mediating role of self-esteem in the connections between cyberbullying and parenting. The study had two goals: (1) to analyze the relation between being a cybervictim and/or cyberaggressor and self-esteem, parents’ acceptance/coercion, and parenting styles and (2) to explore whether self-esteem is a mediator in the relationship between parents’ acceptance/coercion and being a cybervictim/cyberaggressor. Method: The sample comprises 3026 Spanish adolescents (51.5% girls and 48.5% boys) aged 12–18 years (*M_age_* = 14.39; *SD* = 1.69). The study has a cross-sectional design, retrospective ex-post with multiple measurements. Results: (1) cybervictims and cyberaggressors have low self-esteem, and their parents have a low level of involvement/acceptance and a high level of coercion/imposition towards their sons/daughters, (2) participants whose parents were authoritarian obtained significantly lower scores in self-esteem and higher scores in cybervictimization/cyberaggression, whereas those whose parents were indulgent obtained significantly higher scores in self-esteem and lower scores in cybervictimization/cyberaggression, and (3) it was found a mediation of self-esteem in the relationship between the involvement/acceptance of both parents and being a cybervictim, as well as between the father’s coercion/imposition and being a cyberaggressor. Conclusion: An adequate level of self-esteem, high parental acceptance/involvement, and a reasonably low level of coercion/discipline as the parenting style can have very positive effects on the prevention of cyberbullying.

## 1. Introduction

### 1.1. Contextualization

Cyberbullying has been defined as an aggressive behavior carried out using electronic means by a group or an individual repeatedly and over time against a victim who cannot easily defend him or herself [1]. From this perspective, cyberbullying is a systematic abuse of power, and it has been identified as an important problem amongst youth in the last decade [2,3,4,5].

Parents as significant socialization agents explicitly or implicitly transmit social values to their children and have a considerable impact on the development of social behavior. Indeed, past evidence has confirmed that children are more likely to behave pro-socially if parents: (1) are capable to promote children’s safe attachment; (2) provide examples of altruistic behaviors; (3) reinforce and support children’s spontaneous behaviors of help, cooperating, and sharing; (4) transmit to their children that they should not harm other people; (5) make their children restore any harm they have inflicted to others; and (6) adopt a style of inductive discipline on the basis of which they discuss the rules [6,7]. Moreover, the family may be the key aspect as a protection and/or risk factor for the onset of violent behavior [8].

The family is one of the systems that has undergone the most changes due to historical, social, economic, and cultural factors. These changes have led to numerous studies aimed at defining what a family is and the different types of family we find in contemporary society [9,10,11]. The classification of the different current studies of the family is based on the economic and demographic context, the families’ anthropological behavior, the number of their components, and the interactions among them [12].

This family diversity is the reflection of the evolution of society over time, where everyone has a place and plays an important role within a nucleus, with or without kinship. Moreover, it confirms that what has traditionally been known as the “normal family” has ceased to exist, and now, the “normal” is the diversity that makes each family unique, special, and different, and, therefore, not necessarily all types of family, even as socialization agents, represent the culture in which they are registered or transfer the social values of that culture to their children [13].

Research examining protective factors against the development of mental issues has shown the importance of self-esteem in the promotion of personal well-being, mental health, and academic and professional achievement [14,15]. Self-esteem is often defined as a favorable or unfavorable attitude toward the self [16].

Within this contextualization, the present study analyzes the relationship between being a cybervictim or cyberaggressor and self-esteem and the parents’ level of acceptance/coercion, as well as with different parenting styles. Moreover, it explores whether self-esteem is a mediator of the relationship between parents’ acceptance/coercion and being a cybervictim and/or cyberaggressor.

### 1.2. Cyberbullying and Self-Esteem

Relationships between self-esteem and victimization has been extensively studied. Past research has shown that, in childhood, practically any kind of victimization is likely to have a negative impact on self-esteem [17]. Studies exploring the relationship between cyberbullying and self-esteem, in general, shows that the victims have lower levels of self-esteem than people who are not involved in cyberbullying situations [18,19,20,21,22,23]. The findings confirm the deterioration of the self-esteem and self-confidence of students who are the target of continued cyberbullying by their classmates. Further, in another study carried out with cybervictims, 86% of them admitted that suffering cyberbullying had some effect on them, and the most common areas of impact were self-confidence (78%) and self-esteem (70%) [24]. Vilchez [25] has also found a significant inverse correlation between cybervictimization and aggressive-cybervictimization with self-esteem.

The results of studies on cyberaggressors’ self-esteem are contradictory. Some studies state that cyberaggressors have lower levels of self-esteem than people who are not involved in cyberbullying [18,21]; other studies find a negative relation between being a cyberaggressor and self-esteem [22,23], and one study found a positive relation—that is, a good level of self-esteem in cyberaggressors [26]. These discrepant results reveal the need for more research on the topic of study.

Past research has shown that there are factors that make children and adolescents more exposed to cyberbullying victimization, such as social anxiety and social competence [27]. In this last study, the results revealed that increasing worry about others’ evaluation (closely related to self-esteem) is associated with cyberbullying victimization, and likewise, children with poor social skills and difficulties to interact with peers are at greater risk of suffering cyberbullying.

Another variable associated with self-esteem is psychological well-being. The study conducted by Navarro, Ruiz-Oliva, Larrañaga, and Yubero [28] found that cyberbullying victims informed of the worse levels of subjective well-being than children not involved in cyberbullying. Similar results have been found in subsequent research [29,30].

Consequently, research is increasingly analyzing social and emotional resources that facilitate coping with cyberaggression by peers—that is to say, protective factors [31,32]. According to these studies, the increase of social and emotional abilities, such as self-esteem, can have a significant effect on the improvement of health, subjective well-being, and reducing antisocial behaviors, as well as ameliorate the potentially negative effects of cyberbullying. Indeed, research has highlighted the predominant role of self-esteem in predicting adolescents’ psychological adjustment and, by extension, the benefits of developing self-esteem for mental health and well-being among students [33,34].

### 1.3. Cyberbullying and Parenting Styles

In general, studies have revealed that scarce parental attention/control and low family cohesion are associated with cyberbullying (see Buelga, Martínez-Ferrer, and Musitu [35]). The research by Ybarra and Mitchell [36] was one of the first studies to link cyberbullying with family variables. They found that poor family relations (parents’ scarce monitoring, poor emotional bonds, and high level of discipline) were related to more frequent cyberaggression and victimization. Low and Espelage [37] showed that parents’ excessive control is associated with higher levels of the perpetration of cyberbullying. Accordingly, various studies [38,39] have found that low levels of affection and support and the predominance of rejection and aggression towards the children are related to problems of aggressiveness and hostility. Negative family communication is associated with reports of poor parental support, which relates, in turn, with high levels of cyberbullying [40,41], and high family conflict is associated with cybervictimization [42]. Consistent with this, another study [43] showed that two family facets seem to be similarly important against cyberbullying: the perception of family support and perception of rules within the family. While a lack of family support was more associated with cybervictimization, a lack of family rules was more related to cyberaggression. In addition, it has been found that parents’ affection and communication prevent maladaptive behaviors in adolescents [44].

Various studies emphasize that a democratic parenting style, characterized by inductive discipline, affection, and positive control, play a decisive role in the children’s social and behavioral adaptation and in personality aspects such as self-esteem [45,46,47,48]. Some studies have shown that family support is a protective factor for cybervictimization and cyberaggression [49,50]. In Spain, some studies with adolescents have shown that the indulgent style, based primarily on affection (parents’ affective involvement) and not on parental imposition, is associated with better psychological and emotional adjustment [51], socialization [52], and can act as a protective factor for cyberbullying victimization [53].

In contrast, parenting styles characterized by coercive and punitive practices and low affection are related to violence. In general, the authoritarian parenting style is related to greater engagement in proactive and reactive violent behaviors [54]. There is a link between authoritarian and careless styles, characterized by low implication/acceptance, and children participation more in cyberbullying behavior as perpetrators [55]. On the contrary, Dehue, Bolman, Vollink, and Pouwelse [56] found that authoritative parenting styles (characterized by warm and responsive behaviors) were less likely to be related to youth’s cyberbullying behaviors. They also found that youths with neglectful parents were more frequently involved in cyberbullying as aggressors.

Complementarily, past research has found that permissive parenting styles play an important role in parental underestimation of youth’s risky social interactions online [57]. Additionally, research has shown that cybervictims informed of having parents with higher parental stress who used more permissive educational styles, and cyberaggressors informed of having parents with a low level of parental competence [58].

### 1.4. Self-Esteem as a Protective Factor

Despite the research efforts made to understand the psychosocial mechanisms that can help to prevent cyberbullying, little is known about the factors that can improve resilience to overcome cyberbullying. Resilience has been broadly defined as the ability to overcome adverse situations with positive results. Resilience can be fostered by protective factors and inhibited by risk factors. Numerous studies have analyzed the self-esteem and parenting styles of cybervictims and cyberaggressors [59,60], but no prior studies have explored the mediating role of self-esteem in the connections between parental involvement/acceptance and coercion/imposition and cyberbullying—that is, the role that self-esteem may play as a protective factor against cyberbullying and, in consequence, its role in the prevention of violence. 

### 1.5. Goals and Hypotheses

Considering the important prevalence of cyberbullying and its negative consequences [2,61,62], this study analyzes the relationship between being a cybervictim or cyberaggressor and self-esteem and the parents’ level of acceptance/coercion and with parenting styles (negligent, authoritarian, indulgent, and authoritative). It also explores whether self-esteem mediates the relationship between parents’ acceptance/coercion and being a cybervictim or cyberaggressor. 

With these goals in mind, and with reference to the findings of prior works, the study proposes four hypotheses: 

**H1.** 
*Cybervictims will have a low level of self-esteem, whereas the aggressors will have a medium-high level of self-esteem.*


**H2.** 
*The parents of cybervictims and cyberaggressors will show a low level of involvement/acceptance towards their children, and the cyberaggressors’ parents will resort to many coercive and disciplinarian behaviors and impositions as their parenting style.*


**H3.** 
*The authoritarian parenting style will be related to low self-esteem and high scores on the indicators of cyberbullying (cybervictimization, cyberaggression, cyberobservation, and aggressive cybervictimization). However, the indulgent style (combination of high levels of parental warmth and involvement and low levels of strictness and imposition) will be a protective factor against cyberbullying and the factor that most favors self-esteem.*


**H4.** 
*Self-esteem will be a mediator variable of the relation between both the parents’ involvement/acceptance and coercion/imposition and cybervictimization and cyberaggression.*


## 2. Materials and Methods

### 2.1. Participants

The study was conducted with a cross-sectional design. Participants inclusion criteria were: (1) aged 12–18 years old, (2) students in secondary and high schools and (3) enrolled in public or private schools from urban or rural settings in the Basque Country (Northern Spain). The sole exclusion criterion was diagnosis of intellectual or learning disability that could interfere in the responses to the assessment instruments. This yielded an overall sample of 3026 participants, aged 12–18 years old (*M_age_* = 14.39; *SD* = 1.69). Participants were 1469 boys (48.5%) and 1557 girls (51.5%). Participants were enrolled in compulsory secondary education (75.4%) and high school (24.6%) in various public (45.6%) and private (54.4%) schools. Some (19%) participants were in the first year of secondary school (Year 7), 20% were in the second year (Year 8), 17.8% were in the third year (Year 9), and 18.6% were in the final year of secondary school (Year 10). Some (13.6%) participants in high schools were in year 11 and 10.9% were in year 12. Regarding schools’ locations, 79.8% of the participants came from schools located in urban settings, and 20.2% came from schools in rural settings. The distribution of the sample by sex and age is presented in Table 1. The sample size was calculated taking into account the population of students of compulsory secondary education and high school of 101,757 in Basque Country. Using a confidence level of 99%, with a sampling error of 0.024 and a population variance of 0.50, the representative sample was estimated to be 2802 students. To achieve that sample, a stratified, proportional, and randomized sampling technique was used, taking into consideration the proportion of the schools in each province of Basque Country and balancing the following variables (socioeconomic–cultural level and type of school: public–private, urban–rural, secular–religious, etc.).

### 2.2. Procedure

The following phases took place: (1) Directors of the randomly selected schools were contacted, inviting them to participate. (2) An interview was carried out with those directors who agreed to participate in order to describe the project and hand out the informed consent forms for the parents; if the director of the selected center declined the participation, the procedure was repeated with the next center on the list. (3) After reception of the parents’ consent, the research team administered the assessment instruments. The administration took place in two 30-min assessment sessions.

### 2.3. Assessment Instruments

Three scales with adequate psychometric guarantees of reliability and validity were applied to measure the dependent variables.

Cyberbullying. Screening of peer harassment [63,64]. This test assesses 15 cyberbullying behaviors (1 = Offensive/insulting messages; 2 = offensive/insulting calls; 3 = Assaulting, recording and hanging on Internet; 4 = Broadcasting private photos/videos; 5 = Taking photos in dressing rooms, beach, etc. to broadcast; 6 = Anonymous frightening calls; 7 = Blackmailing; 8 = Sexual harassment by mobile/internet; 9 = Identity theft; 10 = Theft of password; 11 = Rigging photos/videos and broadcasting them; 12 = Isolating on social networks; 13 = Blackmailing without broadcasting intimacy; 14 = Death threats; and 15 = Slandering and spreading rumours to discredit someone). The test is made up of 45 items that evaluate the roles performed in cyberbullying: cybervictim, cyberaggressor, and cyberobserver. Participants read each behavior and report the frequency with which they suffered, executed, or observed it during the past year. Each behavior is scored (never = 0, sometimes = 1, several times = 2, and always = 3), and a direct overall score is obtained for each role, respectively. The order of questions for victims, aggressors, and observers was counterbalanced. The test yields percentile scores and 4 indexes: level of cybervictimization, cyberaggression, cyberobservation, and aggressive cybervictimization. Reliability studies confirm a high internal consistency (α = 0.91). The factor analysis yielded a three-factor structure that explained 40.15% of the variance. Studies of convergent validity showed positive correlations between cyberaggression and aggressive conflict resolution, neuroticism, antisocial behavior, school problems, psychopathological disorders, etc. and negative correlations with empathy, responsibility, emotional regulation, and social adjustment.

SES. Self-Esteem Scale [16]. This scale contains 10 items focusing on global feelings of self-appraisal. Participants must read the items and rate them on a 4-point scale ranging from 1 (strongly agree) to 4 (strongly disagree). The reliability of the scale has been broadly documented. For example, McCarthy and Hoge [65] reported consistency coefficients (Cronbach’s alpha) ranging from 0.74 to 0.77 and a test–retest reliability of 0.63 (interval of 7 months) and 0.85 (interval of two weeks). The validity of the scale as a one-dimensional measure of self-esteem has also been confirmed in several studies [16,66]. Cronbach’s alpha reliability in the current sample was satisfactory (α = 0.82). 

ESPA-29. Parental Socialization Scale for Adolescence (in Spanish, Escala de Socialización Parental en la Adolescencia) [67]. This scale measures the parents’ socialization styles. Participants rated the father’s and mother’s actions independently in 29 relevant situations. Of the 29 situations proposed, 16 are negative (items example: “If I am dirty and poorly dressed” or “If I am disobedient”) and 13 are positive (items example: “If I do not skip classes and arrive punctually every day” or “If I pick up and take care of things at home”). The scale allows identifying 4 educational styles of the father and mother along the involvement/acceptance–coercion/imposition continuum: authoritative parents (high involvement/acceptance and high coercion/imposition), indulgent parents (high involvement/acceptance and low coercion/imposition), authoritarian parents (low involvement/acceptance and high coercion/imposition), and neglectful parents (low involvement/acceptance and low coercion/imposition). The results of the factor analysis confirmed the theoretical structure of the bidimensional model satisfactorily [67]. The internal consistency coefficients obtained in the current sample was high (ESPA mother α = 0.91 and ESPA father α = 0.93).

### 2.4. Data Analysis

Firstly, in order to analyze the relations between being a cybervictim and cyberaggressor with the parents’ degree of involvement/acceptance and coercion/imposition, we performed a partial correlational analysis (taking into account the effect of sex, age, and socioeconomic level) of the score in cybervictimization and cyberaggression with self-esteem and with the father’s and mother’s level of involvement/acceptance and coercion/imposition. Subsequently, to determine which parenting style favors higher self-esteem and lower scores on the indicators of cyberbullying (cybervictimization, cyberaggression, cyberobservation, and aggressive cybervictimization), after confirming the basic assumptions, we conducted an analysis of variance with the scores obtained on the three assessment instruments, and, complementarily, we performed a post hoc analysis to compare the groups (Bonferroni) in the four educational styles. Lastly, in order to analyze whether self-esteem mediates between the parents’ involvement/acceptance and coercion/imposition and the probability of becoming a cybervictim or cyberaggressor, we performed a linear regression analysis.

## 3. Results

### 3.1. Relations of Cybervictimization and Cyberaggression with Self-Esteem and with the Degree of Parents’ Involvement/Acceptance and Coercion/Imposition

The results of the partial correlation between cybervictimization and cyberaggression with self-esteem and the level of the father’s and mother’s involvement/acceptance and coercion/imposition are presented in Table 2. As can be observed (see Table 2), significant negative correlations were found between cybervictimization/cyberaggression and the level of both parents’ involvement/acceptance and self-esteem. Significant positive correlations were also found between cybervictimization/cyberaggression and both parents’ coercive/imposing style. Therefore, adolescents and youths who, in the past year, have suffered cyberbullying (cybervictims) and those who have perpetrated it (cyberaggressors) are more likely to have low self-esteem, as well as parents with a low level of acceptance of their children; low involvement in their lives; and a high level of coercion/imposition (rules, limits, punishments, etc.) in the education of their sons and daughters. Nevertheless, we note that the magnitude of the correlations is low in general.

### 3.2. Parenting Styles: Influence on Self-Esteem and on the Level of Cybervictimization, Cyberaggression, Cyberobservation, and Aggressive Cybervictimization

The results of the analysis of variance and the post hoc group comparisons (Bonferroni) of parenting styles, self-esteem, and cyberbullying are presented in Table 3. The data (see Table 3) confirm that the authoritarian educational style is the most harmful, because the adolescents and youths whose parents were authoritarian obtained significantly lower scores in self-esteem and significantly higher scores in all the indicators of cyberbullying (cybervictimization, cyberaggression, cyberobservation, and aggressive cybervictimization). In contrast, the most positive educational style was the indulgent style, because the adolescents whose parents used this educational style had significantly higher scores in self-esteem and lower scores in all the indicators of cyberbullying.

### 3.3. Self-Esteem as a Mediator between Parents’ Acceptance–Coercion and Cybervictimization/Cyberaggression

To analyze whether self-esteem is a mediator between the parents’ involvement/acceptance and coercion/imposition and the probability of becoming a cybervictim or cyberaggressor, we performed a linear regression analysis.

With regard to the mother, on the one hand, we found a perfect or full mediation of self-esteem in the inverse relation between the mother’s acceptance and being a cybervictim (total effect, β = −0.22, *p* = 0.044; partial effect, β = −0.15, *p* = 0.178; Sobel test, Z = −3.37, *p* = 0.001). Therefore, the mother’s low acceptance indirectly predicts being a cybervictim, because self-esteem is a mediator of the magnitude of the relation existing between these two variables. When controlling for the effect of self-esteem, the relation between the mother’s low acceptance and being a cybervictim disappears. The mother’s acceptance is indirectly related to being a cybervictim, because the relation is mediated by self-esteem. That is, even if the mother’s acceptance is low, if the adolescent’s self-esteem is high, such low acceptance does not predict being a cybervictim.

On the other hand, we found a partial mediation of self-esteem in the inverse relation between the mother’s acceptance and being a cyberaggressor (total effect β = −0.48, *p* = 0.001; partial effect, β = −0.44, *p* = 0.000; Sobel test, Z = −4.37, *p* = 0.000). This result shows a partial mediational effect of self-esteem in the relation between the mother’s acceptance and being a cyberaggressor. That is, even though the mother’s acceptance is low, if the adolescent’s self-esteem is high, such low acceptance does not predict being a cyberaggressor. However, self-esteem was not a mediator between the mother’s coercion/imposition and being a cybervictim or cyberaggressor.

With regard to the father, on the one hand, we found partial mediation of self-esteem in the inverse relation between the father’s involvement/acceptance and being a cybervictim (total effect β = −0.26, *p* = 0.003; partial effect, β = −0.21, *p* = 0.023; Sobel test, Z = −3.40, *p* = 0.000). Therefore, even though the father’s acceptance is low, if the adolescent’s self-esteem is high, such low acceptance does not predict being a cybervictim. 

Moreover, we found a partial mediation of self-esteem in the direct relation between the father’s coercion/imposition and being a cyberaggressor (effect total β = 0.49, *p* = 0.001; partial effect β = 0.46, *p* = 0.001; Sobel test, Z = 1.98, *p* = 0.047). Therefore, even if the father’s coercion is high, if the adolescent’s self-esteem is high, such coercion does not predict being a cyberaggressor. However, self-esteem was not a mediator between the father’s involvement/acceptance and being a cyberaggressor or between the father’s coercion/imposition and being a cybervictim.

## 4. Discussion

Firstly, the results showed that adolescents and youths who in the past year have suffered from cyberbullying (cybervictims) and those who have carried it out on others (cyberaggressors) are more likely to have low self-esteem. Therefore, Hypothesis 1 is partially confirmed, because, although we confirmed that cybervictims have low self-esteem, cyberaggressors also show low self-esteem, and we hypothesized a medium-high level. These results point in the same direction as other studies finding low self-esteem in cybervictims [18,19,20,21,22,23,24]. In addition, the results ratify research showing that cyberaggressors have lower levels of self-esteem than people who are not involved in cyberbullying [18,21,22,23]. However, they contradict studies finding that cyberaggressors’ self-esteem was significantly higher [26]. The discrepancies may be explained by the different instruments used to assess self-esteem (some measure global self-esteem, whereas others assess self-esteem in different settings, for example, social, emotional, etc.).

Secondly, the results confirmed that adolescents and youths who, in the past year, have suffered from cyberbullying (cybervictims) and those who have carried it out on others (cyberaggressors) are more likely to have parents with low levels of acceptance of their children; little involvement in their lives; and using high levels of coercion/imposition (rules, limits, punishments, etc.) in the education of their sons and daughters. Therefore, Hypothesis 2 is confirmed.

Thirdly, the authoritarian parenting style was the most harmful, because the adolescents and youths whose parents were authoritarian obtained significantly lower scores in self-esteem and significantly higher scores in all the indicators of cyberbullying (cybervictimization and cyberaggression). In contrast, the most positive educational style was the indulgent style (high involvement/acceptance and low coercion/imposition), because the adolescents whose parents used this educational style had significantly higher scores in self-esteem and lower scores in all the indicators of cyberbullying. Therefore, Hypothesis 3 is confirmed.

The results confirm previous findings that have revealed that authoritarian behavior—high use of physical and verbal coercion, poor family support, hostility, etc.—is related to violent behaviors in children [54]. In contrast, a high level of family support, high parental recognition, and high levels of involvement and warmth in the children’s lives are related to lower levels of violent behavior in these children. These results are in line with other studies finding that parents’ scarce supervision, poor emotional bonds, and high level of discipline are related to cyberbullying [36], whereas parents’ excessive control is associated with cyberaggression [37], and a good relation with one’s parents, based on the positive expression of feelings, is a protective factor against cybervictimization [43,44,48,51,52]. The results also ratify other studies showing that adolescents with authoritative parents were less likely to engage in online cyberbullying [56]. Small differences in the relationship between parenting styles and cyberbullying (indulgent versus authoritative) found in different studies can be explained in terms of cultural differences. Consequently, more research about more positive parenting styles in different cultures is needed.

Lastly, the results of the mediational analyses found: (1) mediation of self-esteem in the inverse relationship between the mother’s involvement/acceptance and being a cybervictim or cyberaggressor (even if the mother’s acceptance is low, this does not predict being a cybervictim or a cyberaggressor if the adolescent’s self-esteem is high) and (2) mediation of self-esteem in the inverse relation between the father’s involvement/acceptance and being a cybervictim, as well as the in direct relationship between the father’s coercion/imposition and being a cyberaggressor (even if the father’s acceptance is low, this does not predict being a cybervictim if the adolescent’s self-esteem is high, and even if the father’s coercion is high, this does not predict being a cyberaggressor if the adolescent’s self-esteem is high). Summing up, self-esteem is a mediator in the relationship between the involvement/acceptance of both parents and being a cybervictim and between the mother’s involvement/acceptance and being a cyberaggressor, as well as between the father’s coercion/imposition and being a cyberaggressor. Therefore, Hypothesis 4 is partially confirmed.

### Limitations and Practical Implications

Limitations should be noted when interpreting the results of the current study. Among others, the use of self-reports with the implied bias of social desirability; therefore, for future studies, we recommend administering the assessment instruments concerning educational practices to the parents, as this would allow contrasting the results obtained in this study. Additionally, the cross-sectional design employed limits our capacity to make causal inferences regarding the relationships found in the study. Longitudinal studies are needed to analyze the causal relation that links parental styles and indicators of cyberbullying.

Despite these limitations, the study has important practical implications for school health professionals The results suggest the importance of implementing at school programs to promote self-esteem particularly, along with other factors of socioemotional development (e.g., prosocial behavior, assertive communication, empathy, etc.), due to its efficacy in decreasing violence and anti-cyberbullying programs that also stimulate transversally socioemotional development factors [68,69,70,71,72,73].

Another practical implication is related to the family as a socialization context. The results suggest the importance of working with the parents to foster their high involvement in and acceptance of their children, with a medium level of coercion/imposition, which would help to prevent cyberbullying. Additionally, the results allow us to suggest the importance of training spaces, where the parents can learn positive educational patterns that inhibit violent behavior in all its modalities for the development of their children. In these training contexts, parents could learn: (1) how to stimulate positive and realistic self-esteem in their children; (2) how to deal constructively with conflicts with their children; (3) to be aware of the importance of their behavior as a model, as well as the role of the contingencies they provide for their children’s behaviors; (4) positive ways of communication and dialogue with their children to know about diverse aspects of their lives (for example, their behavior on Internet and with their peers); and (5) the negative influence of observation of violent behaviors (for example, on the Internet, in videogames, etc.).

However, it should be noted that the results of this study are limited by the educational styles analyzed, which are conditioned in part by the measuring instrument used to assess the influence of the family (ESPA). From the most current approaches, which recognize plurality in the exercise of parenting, it is more appropriate to refer to parental practices or competencies that promote the children’s adequate development and adjustment. There is growing evidence that the typological approach that indicates the democratic style as the only valid style is restrictive, because, in certain contexts and/or with certain minors, other parental educational practices may be more appropriate. It is necessary to take into account the ecology of parentality that helps us to understand that there is an important diversity and that there are certain basic parental competencies that should be present but that should always be adapted to the specific characteristics and needs of each child and the specific context of each family [74].

Therefore, in the context of this debate, it is important to include the current approach of positive parenting, which is a concept that refers to parental behavior based on the children’s best interests, which cares for and develops their abilities, is not violent and offers recognition and guidance that include the establishment of limits that allow the child’s full development [74], emphasizing the need to implement effective parental support interventions carried out by professionals registered in educational and family support services and coordinating municipal, autonomic, and ministerial services (Ministry of Health, Social Services, and Equality, etc.).

With regard to the principles of positive parenting, and in line with the recommendations of the Council of Europe Committee of Ministers, Rodrigo, Máiquez, and Martín [75,76] proposed six principles of action for positive parenting, which have been shown to enhance children’s adequate development, promoting their physical and mental well-being. These principles of positive parenting are: (1) Warm, protective, and stable affective bonds to make the children feel accepted and loved (this implies the continued strengthening of family ties throughout their development, changing the forms of expression of affection with age); (2) A structured environment, providing a model, guidance, and supervision for the children to learn the rules and values; (3) Stimulation and support for daily and school learning to promote their motivation and capacities; (4) Recognition of the children’s value, showing interest in their world, validating their experiences, becoming involved in their concerns, and responding to their needs; (5) Training the children, enhancing their self-perception as active agents who are competent and capable of changing things and influencing others; and (6) Nonviolent education, excluding any form of degrading physical or psychological punishment. 

However, the task of parenting does not depend exclusively on the parents’ characteristics. It is exercised within an ecological space where the quality depends on three types of factors: (1) the psychosocial context, (2) the evolving and educational needs of the child, and (3) the parents’ ability to exercise positive parenting [77,78]. Parenting is not exercised in a vacuum but in various ecologies or psychosocial environments that facilitate or hinder the exercise of such responsibility. For more information on the influence of context, see Arranz [79,80] and Arranz, Oliva, Sánchez, Olabarrieta, and Richard [81].

As Manzano et al. [78] emphasized, the goal of positive parentality policies is to get mothers and fathers to train for their teaching roles. In addition, it is important to take into account the distinction between the parenting dimensions and/or the family context variables and the strictly parental competences. It is important to note that not all intervention actions in positive parenthood focus on the families’ acquisition of parental competencies. There are aspects, such as the economic and educational levels of the families or the neighborhood where they reside, that influence the quality of the family context without being, per se, parental competencies. In short, it is essential to apply an ecological design [82,83] that includes positive parentality variables and positive parentality competencies. 

In this context, the proposal of the “Currículo Óptimo de Parentalidad Positiva” (COPP; Optimal Positive Parentality Curriculum) identifies three parental competencies that include diverse variables to consider during infancy, childhood, and adolescence: (1) cognitive and linguistic development (including, among others, learning stimulation, stimulation of cognitive development, stimulation of play, stimulation of the theory of mind,); (2) socioemotional development (including, among others, stimulation of social maturity, existence of rules and limits, promotion of critical thinking, stimulation of autonomy, stimulation of self-esteem, emotion regulation, adequate parental control, authorized style of conflict resolution, promotion of sexual education, and transmission of universal ethical values); and (3) social context and physical environment (including, among others, the father’s involvement, low levels of family stress, stable and effective social support network, quality institutional resources, family sharing of free time, media exposure supervision, health education, and adequate family housing). In addition, the COPP includes some transversal variables, such as appropriate socioeconomic status, educational flexibility, knowledge of the phases of psychological development, implicit environmental theories, active listening and empathy, assertiveness, and interaction routines. This proposal includes the parents’ competencies that must be promoted in order to achieve an exercise of positive parenting that is effective as a factor in protection against cyberbullying [77,78].

## 5. Conclusions

The results obtained suggest that an adequate level of self-esteem, high parental acceptance/involvement, and a reasonably low level of coercion/discipline as the parenting style can have very positive effects on the decrease of violent peer behavior and on the prevention of cyberbullying. The authoritarian parental style is related to cyberbullying and low self-esteem, while the indulgent style is a protector factor against cyberbullying and favors self-esteem. Additionally, self-esteem is a mediator that might help to explain the psychological mechanisms that underline the link between parental styles and cyberbullying involvement. The current study makes two specific contributions: (1) It supports the importance of self-esteem as a protective factor against cyberbullying (cybervictimization/cyberaggression), and this may indicate that adequate prevention and treatment policies should focus on adolescents’ sense of being a good person, according to their own criteria of worth. School programs and interventions fostering social emotional learning are already being designed and implemented worldwide to prevent cyberbullying [84]. (2) It emphasizes the important role of parents in preventing cyberbullying through their educational behaviors with their children. Intervention programs against bullying/cyberbullying should extend their focus beyond schools to include families, and they should start before children enter school [85]. The results suggest promoting the self-esteem of children and adolescents as a way to neutralize the relationship between the parent’s involvement/acceptance and coercion/imposition and the possibility of becoming a cybervictim or cyberaggressor.

## Figures and Tables

**Table 1 children-09-01795-t001:** Description of the sample: frequency and percentage of boys and girls in the three age groups.

	12–13 Years	14–15 Years	16–18 Years	Total
Boys	543 (51.2%)	536 (49%)	390 (44.8%)	1469 (48.5%)
Girls	518 (48.8%)	558 (51%)	481 (55.2%)	1557 (51.5%)
Total	1061 (100%)	1094 (100%)	871 (100%)	3026 (100%)

**Table 2 children-09-01795-t002:** Partial correlations of cybervictimization and cyberaggression with self-esteem and with parents’ acceptance–coercion.

	Cybervictimization	Cyberaggression
Self-esteem	−0.12 ***	−0.07 ***
Mother’s involvement/acceptance	−0.05 *	−0.09 ***
Mother’s coercion/imposition	0.09 ***	0.07 **
Father’s involvement/acceptance	−0.08 ***	−0.08 ***
Father’s coercion/imposition	0.10 ***	0.08 ***

* *p* < 0.05. ** *p* < 0.01. *** *p* < 0.001.

**Table 3 children-09-01795-t003:** Means, standard deviations, analysis of variance, and post hoc (Bonferroni) analysis of the parenting styles with self-esteem and with indicators of cyberbullying.

Mother										
	Negligent		Authoritarian		Indulgent		Authoritative		F(3, 3022)	Post Hoc
	M (SD)		M (SD)		M (SD)		M (SD)			
Self-esteem	29.50 (4.83)		28.33 (5.26)		30.21 (4.84)		30.11 (5.57)		8.71 ***	2 < 3,4
Cybervictimization	0.73 (1.95)		1.03 (2.44)		0.59 (1.46)		0.81 (1.84)		3.44 *	2 > 3
Cyberaggression	0.41 (1.52)		0.81 (3.57)		0.26 (1.92)		0.34 (1.28)		4.66 **	2 > 3,4
Cyberobservation	2.88 (3.16)		4.94 (5.83)		2.80 (3.87)		3.46 (4.23)		15.85 ***	2 > 1,3,4; 4 > 3
Aggressive-cybervictimization	1.14 (3.18)		1.84 (4.96)		0.85 (3.02)		1.14 (2.64)		5.23 ***	2 > 3,4
Father										
	Negligent		Authoritarian		Indulgent		Authoritative		F(3, 3022)	Post Hoc
	M	SD	M	SD	M	SD	M	SD		
Self-esteem	29.70 (4.81)		28.56 (5.04)		30.31 (5.10)		30.26 (5.51)		8.20 ***	2 < 1,3,4
Cybervictimization	0.79 (1.95)		1.03 (2.41)		0.46 (1.09)		0.80 (1.81)		5.91 ***	2,4 > 3
Cyberaggression	0.43 (2.36)		0.69 (3.21)		0.20 (0.92)		0.33 (1.38)		3.63 *	2 > 3
Cyberobservation	3.14 (4.18)		4.64 (5.16)		2.83 (3.95)		3.49 (4.22)		10.75 ***	2 > 1,3,4
Aggressive-cybervictimization	1.22 (3.88)		1.72 (4.52)		0.66 (1.60)		1.13 (2.78)		6.21 ***	2 > 3

Notes. * *p* < 0.05. ** *p* < 0.01. *** *p* < 0.001. (1) = Negligent, (2) = Authoritarian, (3) = Indulgent, and (4) = Authoritative.

## Data Availability

Data are not publicly available.

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
