# Peer review of "Parenting Styles and Self-Esteem in Adolescent Cybervictims and Cyberaggressors: Self-Esteem as a Mediator Variable"

_children, 2022, doi:10.3390/children9121795_

Round 1

Reviewer 1 Report

The topic dealt with is of great interest and extremely topical, cyberbullying represent a serious problem of public health.
The study described in this manuscript was carried out with scientific rigor and the modalities are clearly illustrated. The results are presented clearly and the discussion is conducted critically and comprehensively.
The quality of tables is satisfactory. The references are relevants and I think that implementation would be desirable, especially when considering possible prevention strategies involving parents and schools, in order to deepen the paragraph 1.3 “cyberbullying and parental styles”. The article certainly deserves to be published, with small revisions in the bibliography.

Here few examples of literature that can be added in references:

Gabrielli, S.; Rizzi, S.; Carbone, S.; Piras, E.M. School Interventions for Bullying–Cyberbullying Prevention in Adolescents: Insights from the UPRIGHT and CREEP Projects. Int. J. Environ. Res. Public Health 202118, 11697. https://doi.org/10.3390/ijerph182111697

Mameli, C.; Menabò, L.; Brighi, A.; Menin, D.; Culbert, C.; Hamilton, J.; Scheithauer, H.; Smith, P.K.; Völlink, T.; Willems, R.A.; Purdy, N.; Guarini, A. Stay Safe and Strong: Characteristics, Roles and Emotions of Student-Produced Comics Related to Cyberbullying. Int. J. Environ. Res. Public Health 202219, 8776. https://doi.org/10.3390/ijerph19148776

Tozzo, P.; Cuman, O.; Moratto, E.; Caenazzo, L. Family and Educational Strategies for Cyberbullying Prevention: A Systematic Review. Int. J. Environ. Res. Public Health 202219, 10452. https://doi.org/10.3390/ijerph191610452

Author Response

First, we are extremely grateful for the comments from the reviewers, which we believe have contributed to significant improvements in the paper.

We have commented on each piece of feedback separately and we have attached an updated version of our paper after revisions.

Changes introduced in the manuscript haven been tracked in red.

Reviewer 1

The topic dealt with is of great interest and extremely topical, cyberbullying represent a serious problem of public health.
The study described in this manuscript was carried out with scientific rigor and the modalities are clearly illustrated. The results are presented clearly and the discussion is conducted critically and comprehensively.
The quality of tables is satisfactory. The references are relevants and I think that implementation would be desirable, especially when considering possible prevention strategies involving parents and schools, in order to deepen the paragraph 1.3 “cyberbullying and parental styles”. The article certainly deserves to be published, with small revisions in the bibliography.

Here few examples of literature that can be added in references:

Gabrielli, S.; Rizzi, S.; Carbone, S.; Piras, E.M. School Interventions for Bullying–Cyberbullying Prevention in Adolescents: Insights from the UPRIGHT and CREEP Projects. Int. J. Environ. Res. Public Health 202118, 11697. https://doi.org/10.3390/ijerph182111697

Mameli, C.; Menabò, L.; Brighi, A.; Menin, D.; Culbert, C.; Hamilton, J.; Scheithauer, H.; Smith, P.K.; Völlink, T.; Willems, R.A.; Purdy, N.; Guarini, A. Stay Safe and Strong: Characteristics, Roles and Emotions of Student-Produced Comics Related to Cyberbullying. Int. J. Environ. Res. Public Health 202219, 8776. https://doi.org/10.3390/ijerph19148776

Tozzo, P.; Cuman, O.; Moratto, E.; Caenazzo, L. Family and Educational Strategies for Cyberbullying Prevention: A Systematic Review. Int. J. Environ. Res. Public Health 202219, 10452. https://doi.org/10.3390/ijerph191610452

Authors’ response: thank you very much for your kind review and to point out some relevant studies related with our manuscript. We have reviewed the references suggested, as well as other, and we have update our citations and reference list accordingly.

Reviewer 2 Report

Title: Parenting styles and self-esteem in cybervictims and cyberaggressors: self-esteem as a mediator variable

 Summary

In this longitudinal study, the authors investigated the relation between being a cybervictim and/or cyberaggressor and self-esteem, parents' acceptance/coercion, and parenting styles as well as the mediator role of self-esteem in the relationship between parents' acceptance/coercion and being a cybervictim/cyberaggressor. Participants were 3026 Spanish adolescents aged between 12 -18. The results indicated that cybervictims and cyberaggressors adolescents have low self-esteem, and their parents have a low level of involvement/acceptance and a high level of coercion/imposition towards their sons/daughters.  The results also showed adolescents whose parents were authoritarian obtained significantly lower scores in self-esteem and higher scores in cybervictimization/cyberaggression, whereas those whose parents were indulgent obtained significantly higher scores in self-esteem and lower scores in cybervictimization/cyberaggression. The authors also reported the mediator role of self-esteem in the relationship between the involvement/acceptance of both parents and being a cybervictim, as well as between the father's coercion/imposition and being a cyberaggressors.

 Points and suggestions

-          Since this research was conducted on adolescent, please refer to adolescents in the title as well as in the abstract.

-          Please add a small introduction in the first line of the abstract.

-          Please add the mean (SD) age to the abstract.

-          Please add the gender of the participants in the abstract, how many were boys and or girls?

-          Please speak about the research type in the abstract.

-          Please add a better conclusion in the last line of the abstract.

-          The introduction seems too long. It is better to write more briefly.

-          Please add more information regarding the participants in the Participants section.

-          Please add inclusion and exclusion criteria to the method section.

-          Regarding the instrument used in this study, please also cite articles that have used this instrument in adolescents. Most of the citations you wrote are for adults.

-          The discussion also seems too long. It is better to write more briefly.

Author Response

First, we are extremely grateful for the comments from the reviewers, which we believe have contributed to significant improvements in the paper.

We have commented on each piece of feedback separately and we have attached an updated version of our paper after revisions.

Changes introduced in the manuscript haven been tracked in red.

Reviewer 2

1) Since this research was conducted on adolescent, please refer to adolescents in the title as well as in the abstract.

Authors’ response: thank you for this suggestion. We have now referred to adolescents in the title and the abstract.

2) Please add a small introduction in the first line of the abstract.

Authors’ response: thank you. The abstract is now introduced by the sentence: Family relationships and self-esteem are relevant variables into the understanding of cyberbullying. However, little is known about the mediating role of self-esteem in the connections between cyberbullying and parenting.

3) Please add the mean (SD) age to the abstract. Please add the gender of the participants in the abstract, how many were boys and or girls?

Authors’ response: thank you. Mean age and SD is now reported in the abstract. Gender has also been reported in the abstract.

4) Please speak about the research type in the abstract.

Authors’ response: thank you. Research type is now indicated in the abstract.

5) Please add a better conclusion in the last line of the abstract.

Authors’ response: thank you. A new line has been included in the abstract regarding conclusion of the study: An adequate level of self-esteem, high parental acceptance/involvement, and a reasonably low level of coercion/discipline as the parenting style can have very positive effects on the prevention of cyberbullying

6) The introduction seems too long. It is better to write more briefly.

Authors’ response: thank you for this suggestion. We have made several edits to shorten the introduction.

7) Please add more information regarding the participants in the Participants section.

Authors’ response: thank you. We have now included information regarding participants distribution in secondary and high school years and urban and rural location of the school involved in the study.

8) Please add inclusion and exclusion criteria to the method section.

Authors’ response: thank you for this suggestion. The inclusion and exclusion criteria have been described at the beginning of the participants subsection.

9) Regarding the instrument used in this study, please also cite articles that have used this instrument in adolescents. Most of the citations you wrote are for adults.

Authors’ response: thank you for pointing this out. We have changed reference number 61 to include a study validating the unidimensional scale in a sample of adolescents.

 10) The discussion also seems too long. It is better to write more briefly.

Authors’ response: thank you for this suggestion. We have made several edits to shorten the discussion.

Round 2

Reviewer 2 Report

The authors have successfully addressed my comments, so I would recommend it for publication.